# Improving Generalization in Meta Reinforcement Learning using Learned Objectives

**Louis Kirsch, Sjoerd van Steenkiste, Jürgen Schmidhuber**
The Swiss AI Lab IDSIA, USI, SUPSI
{louis, sjoerd, juergen}@idsia.ch

## Abstract

Biological evolution has distilled the experiences of many learners into the general learning algorithms of humans. Our novel meta reinforcement learning algorithm *MetaGenRL* is inspired by this process. *MetaGenRL* distills the experiences of many complex agents to meta-learn a low-complexity neural objective function that decides how future individuals will learn. Unlike recent meta-RL algorithms, *MetaGenRL* can generalize to new environments that are entirely different from those used for meta-training. In some cases, it even outperforms human-engineered RL algorithms. *MetaGenRL* uses off-policy second-order gradients during meta-training that greatly increase its sample efficiency.

## 1 Introduction

The process of evolution has equipped humans with incredibly general learning algorithms. They enable us to solve a wide range of problems, even in the absence of a large number of related prior experiences. The algorithms that give rise to these capabilities are the result of *distilling* the collective experiences of many learners throughout the course of natural evolution. By essentially *learning from learning experiences* in this way, the resulting knowledge can be compactly encoded in the genetic code of an individual to give rise to the general learning capabilities that we observe today.

In contrast, Reinforcement Learning (RL) in artificial agents rarely proceeds in this way. The learning rules that are used to train agents are the result of years of human engineering and design, (e.g. Williams (1992); Wierstra et al. (2008); Mnih et al. (2013); Lillicrap et al. (2016); Schulman et al. (2015a)). Correspondingly, artificial agents are inherently limited by the ability of the designer to incorporate the right inductive biases in order to learn from previous experiences.

Several works have proposed an alternative framework based on *meta reinforcement learning* (Schmidhuber, 1994; Wang et al., 2016; Duan et al., 2016; Finn et al., 2017; Houthooft et al., 2018; Clune, 2019). Meta-RL distinguishes between learning to act in the environment (the reinforcement learning problem) and learning to learn (the meta-learning problem). Hence, learning itself is now a learning problem, which in principle allows one to leverage prior learning experiences to meta-learn *general* learning rules that surpass human-engineered alternatives. However, while prior work found that learning rules could be meta-learned that generalize to slightly different environments or goals (Finn et al., 2017; Plappert et al., 2018; Houthooft et al., 2018), generalization to *entirely different* environments remains an open problem.

In this paper we present *MetaGenRL*[1], a novel meta reinforcement learning algorithm that meta-learns learning rules that generalize to entirely different environments. MetaGenRL is inspired by the process of natural evolution as it distills the experiences of many agents into the parameters of an objective function that decides how future individuals will learn. Similar to Evolved Policy Gradients (EPG; Houthooft et al. (2018)), it meta-learns low complexity neural objective functions that can be used to train complex agents with many parameters. However, unlike EPG, it is able to meta-learn using second-order gradients, which offers several advantages as we will demonstrate.

We evaluate MetaGenRL on a variety of continuous control tasks and compare to RL[2] (Wang et al., 2016; Duan et al., 2016) and EPG in addition to several human engineered learning algorithms.

---

[1]Code is available at http://louiskirsch.com/code/metagenrl

Compared to RL$^2$ we find that MetaGenRL does not overfit and is able to train randomly initialized agents using meta-learned learning rules on *entirely different* environments. Compared to EPG we find that MetaGenRL is more sample efficient, and outperforms significantly under a fixed budget of environment interactions. The results of an ablation study and additional analysis provide further insight into the benefits of our approach.

## 2 PRELIMINARIES

**Notation**   We consider the standard MDP Reinforcement Learning setting defined by a tuple $e = (S, A, P, \rho_0, r, \gamma, T)$ consisting of states $S$, actions $A$, the transition probability distribution $P : S \times A \times S \to \mathbb{R}_+$, an initial state distribution $\rho_0 : S \to \mathbb{R}_+$, the reward function $r : S \times A \to [-R_{max}, R_{max}]$, a discount factor $\gamma$, and the episode length $T$. The objective for the probabilistic policy $\pi_\phi : S \times A \to \mathbb{R}_+$ parameterized by $\phi$ is to maximize the expected discounted return:

$$\mathbb{E}_\tau[\sum_{t=0}^{T-1} \gamma^t r_t], \text{ where } s_0 \sim \rho_0(s_0),\ a_t \sim \pi_\phi(a_t|s_t),\ s_{t+1} \sim P(s_{t+1}|s_t, a_t), r_t = r(s_t, a_t), \quad (1)$$

with $\tau = (s_0, a_0, r_0, s_1, ..., s_{T-1}, a_{T-1}, r_{T-1})$.

**Human Engineered Gradient Estimators**   A popular gradient-based approach to maximizing Equation 1 is REINFORCE (Williams, 1992). It directly differentiates Equation 1 with respect to $\phi$ using the likelihood ratio trick to derive gradient estimates of the form:

$$\nabla_\phi \mathbb{E}_\tau[L_{REINF}(\tau, \pi_\phi)] := \mathbb{E}_\tau[\nabla_\phi \sum_{t=0}^{T-1} \log \pi_\phi(a_t|s_t) \cdot \sum_{t'=t}^{T-1} \gamma^{t'-t} r_{t'})]. \quad (2)$$

Although this basic estimator is rarely used in practice, it has become a building block for an entire class of policy-gradient algorithms of this form. For example, a popular extension from Schulman et al. (2015b) combines REINFORCE with a Generalized Advantage Estimate (GAE) to yield the following policy gradient estimator:

$$\nabla_\phi \mathbb{E}_\tau[L_{GAE}(\tau, \pi_\phi, V)] := \mathbb{E}_\tau[\nabla_\phi \sum_{t=0}^{T-1} \log \pi_\phi(a_t|s_t) \cdot A(\tau, V, t)]. \quad (3)$$

where $A(\tau, V, t)$ is the GAE and $V : S \to \mathbb{R}$ is a value function estimate. Several recent other extensions include TRPO (Schulman et al., 2015a), which discourages bad policy updates using trust regions and iterative off-policy updates, or PPO (Schulman et al., 2017), which offers similar benefits using only first order approximations.

**Parametrized Objective Functions**   In this work we note that many of these human engineered policy gradient estimators can be viewed as specific implementations of a general objective function $L$ that is differentiated with respect to the policy parameters:

$$\nabla_\phi \mathbb{E}_\tau[L(\tau, \pi_\phi, V)]. \quad (4)$$

Hence, it becomes natural to consider a generic parametrization of $L$ that, for various choices of parameters $\alpha$, recovers some of these estimators. In this paper, we will consider *neural objective functions* where $L_\alpha$ is implemented by a neural network. Our goal is then to optimize the parameters $\alpha$ of this neural network in order to give rise to a new learning algorithm that best maximizes Equation 1 on an entire class of (different) environments.

## 3 META-LEARNING NEURAL OBJECTIVES

In this work we propose *MetaGenRL*, a novel meta reinforcement learning algorithm that meta-learns neural objective functions of the form $L_\alpha(\tau, \pi_\phi, V)$. MetaGenRL makes use of value functions and second-order gradients, which makes it more sample efficient compared to prior work (Duan et al., 2016; Wang et al., 2016; Houthooft et al., 2018). More so, as we will demonstrate, MetaGenRL meta-learns objective functions that generalize to vastly different environments.

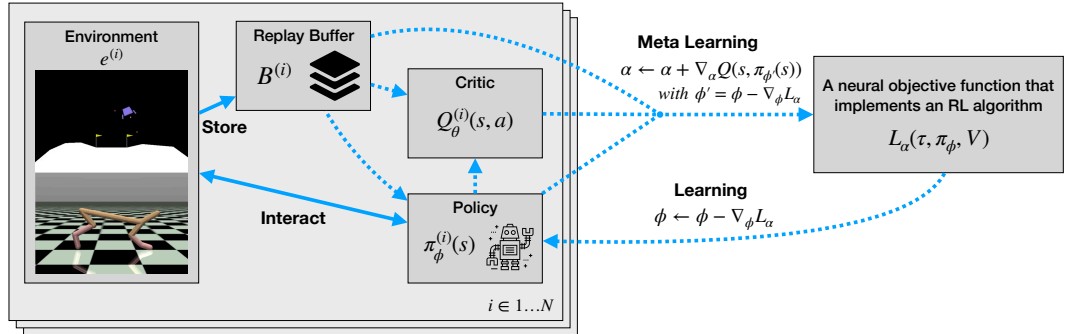

Figure 1: A schematic of *MetaGenRL*. On the left a population of agents ($i \in 1, \ldots, N$), where each member consist of a critic $Q_\theta^{(i)}$ and a policy $\pi_\phi^{(i)}$ that interact with a particular environment $e^{(i)}$ and store collected data in a corresponding replay buffer $B^{(i)}$. On the right a meta-learned neural objective function $L_\alpha$ that is shared across the population. Learning (dotted arrows) proceeds as follows: Each policy is updated by differentiating $L_\alpha$, while the critic is updated using the usual TD-error (not shown). $L_\alpha$ is meta-learned by computing second-order gradients that can be obtained by differentiating through the critic.

Our key insight is that a differentiable critic $Q_\theta : S \times A \to \mathbb{R}$ can be used to measure the effect of locally changing the objective function parameters $\alpha$ based on the quality of the corresponding policy gradients. This enables a population of agents to use and improve a single parameterized objective function $L_\alpha$ through interacting with a set of (potentially different) environments. During evaluation (meta-test time), the meta-learned objective function can then be used to train a randomly initialized RL agent in a new environment.

### 3.1 FROM DDPG TO GRADIENT-BASED META-LEARNING OF NEURAL OBJECTIVES

We will formally introduce MetaGenRL as an extension of the DDPG actor-critic framework (Silver et al., 2014; Lillicrap et al., 2016). In DDPG, a parameterized critic of the form $Q_\theta : S \times A \to \mathbb{R}$ transforms the non-differentiable RL reward maximization problem into a myopic value maximization problem for any $s_t \in S$. This is done by alternating between optimization of the critic $Q_\theta$ and the (here deterministic) policy $\pi_\phi$. The critic is trained to minimize the TD-error by following:

$$\nabla_\theta \sum_{(s_t,a_t,r_t,s_{t+1})} (Q_\theta(s_t, a_t) - y_t)^2, \text{ where } y_t = r_t + \gamma \cdot Q_\theta(s_{t+1}, \pi_\phi(s_{t+1})), \quad (5)$$

and the dependence of $y_t$ on the parameter vector $\theta$ is ignored. The policy $\pi_\phi$ is improved to increase the expected return from arbitrary states by following the gradient $\nabla_\phi \sum_{s_t} Q_\theta(s_t, \pi_\phi(s_t))$. Both gradients can be computed entirely off-policy by sampling trajectories from a replay buffer.

MetaGenRL builds on this idea of differentiating the critic $Q_\theta$ with respect to the policy parameters. It incorporates a parameterized objective function $L_\alpha$ that is used to improve the policy (i.e. by following the gradient $\nabla_\phi L_\alpha$), which adds one extra level of indirection: The critic $Q_\theta$ improves $L_\alpha$, while $L_\alpha$ improves the policy $\pi_\phi$. By first differentiating with respect to the objective function parameters $\alpha$, and then with respect to the policy parameters $\phi$, the critic can be used to measure the effect of updating $\pi_\phi$ using $L_\alpha$ on the estimated return[2]:

$$\nabla_\alpha Q_\theta(s_t, \pi_{\phi'}(s_t)), \text{ where } \phi' = \phi - \nabla_\phi L_\alpha(\tau, x(\phi), V). \quad (6)$$

This constitutes a type of second order gradient $\nabla_\alpha \nabla_\phi$ that can be used to meta-train $L_\alpha$ to provide better updates to the policy parameters in the future. In practice we will use batching to optimize Equation 6 over multiple trajectories $\tau$.

Similarly to the policy-gradient estimators from Section 2, the objective function $L_\alpha(\tau, x(\phi), V)$ receives as inputs an episode trajectory $\tau = (s_{0:T-1}, a_{0:T-1}, r_{0:T-1})$, the value function estimates

---

[2]In case of a probabilistic policy $\pi_\phi(a_t|s_t)$ the following becomes an expectation under $\pi_\phi$ and a reparameterizable form is required (Williams, 1988; Kingma & Welling, 2014; Rezende et al., 2014). Here we focus on learning deterministic target policies.

---

**Algorithm 1** MetaGenRL: Meta-Training

---

**Require:** $p(e)$ a distribution of environments
    $P \Leftarrow \{(e_1 \sim p(e), \phi_1, \theta_1, B_1 \leftarrow \varnothing), \ldots\}$           ▷ Randomly initialize population of agents
    Randomly initialize objective function $L_\alpha$
    **while** $L_\alpha$ has not converged **do**
        **for** $e, \phi, \theta, B \in P$ **do**                           ▷ For each agent $i$ in parallel
            **if** extend replay buffer $B$ **then**
                Extend $B$ using $\pi_\phi$ in $e$
            Sample trajectories from $B$
            Update critic $Q_\theta$ using TD-error
            Update policy by following $\nabla_\phi L_\alpha$
            Compute objective function gradient $\Delta_i$ for agent $i$ according to Equation 6
        Sum gradients $\sum_i \Delta_i$ to update $L_\alpha$

---

$V$, and an auxiliary input $x(\phi)$ (previously $\pi_\phi$) that can be differentiated with respect to the policy parameters. The latter is critical to be able to differentiate with respect to $\phi$ and in the simplest case it consists of the action as predicted by the policy. While Equation 6 is used for meta-learning $L_\alpha$, the objective function $L_\alpha$ itself is used for policy learning by following $\nabla_\phi L_\alpha(\tau, x(\phi), V)$. See Figure 1 for an overview. MetaGenRL consists of two phases: During meta-training, we alternate between critic updates, objective function updates, and policy updates to meta-learn an objective function $L_\alpha$ as described in Algorithm 1. During meta-testing in Algorithm 2, we take the learned objective function $L_\alpha$ and keep it fixed while training a randomly initialized policy in a new environment to assess its performance.

We note that the inputs to $L_\alpha$ are sampled from a replay buffer rather than solely using on-policy data. If $L_\alpha$ were to represent a REINFORCE-type objective then it would mean that differentiating $L_\alpha$ yields *biased* policy gradient estimates. In our experiments we will find that the gradients from $L_\alpha$ work much better in comparison to a biased off-policy REINFORCE algorithm, and to an importance-sampled unbiased REINFORCE algorithm, while also improving over the popular on-policy REINFORCE and PPO algorithms.

### 3.2   Parametrizing the Objective Function

We will implement $L_\alpha$ using an LSTM (Gers et al., 2000; Hochreiter & Schmidhuber, 1997) that iterates over $\tau$ in reverse order and depends on the current policy action $\pi_\phi(s_t)$ (see Figure 2). At every time-step $L_\alpha$ receives the reward $r_t$, taken action $a_t$, predicted action by the current policy $\pi_\phi(s_t)$, the time $t$, and value function estimates $V_t, V_{t+1}$[3]. At each step the LSTM outputs the objective value $l_t$, all of which are summed to yield a single scalar output value that can be differentiated with respect to $\phi$. In order to accommodate varying action dimensionalities across different environments, both $\pi_\phi(s_t)$ and $a_t$ are first convolved and then averaged to obtain an action embedding that does not depend on the action dimensionality. Additional details, including suggestions for more expressive alternatives are available in Appendix B.

By presenting the trajectory in reverse order to the LSTM (and $L_\alpha$ correspondingly), it is able to assign credit to an action $a_t$ based on its *future* impact on the reward, similar to policy gradient estimators. More so, as a general function approximator using these inputs, the LSTM is in principle able to learn different variance and bias reduction techniques, akin to advantage estimates, generalized advantage estimates, or importance weights[4]. Due to these properties, we expect the class of objective functions that is supported to somewhat relate to a REINFORCE (Williams, 1992) estimator that uses generalized advantage estimation (Schulman et al., 2015b).

---

[3]The value estimates are derived from the Q-function, i.e. $V_t = Q_\theta(s_t, \pi_\phi(s_t))$, and are treated as a constant input. Hence, the gradient $\nabla_\phi L_\alpha$ can not flow backwards through $Q_\theta$, which ensures that $L_\alpha$ can not naively learn to implement a DDPG-like objective function.

[4]We note that in practice it is is difficult to assess whether the meta-learned object function incorporates bias / variance reduction techniques, especially because MetaGenRL is unlikely to recover *known* techniques.

**Algorithm 2** MetaGenRL: Meta-Testing

**Require:** A test environment $e$, and an
    objective function $L_\alpha$
    Randomly initialize $\pi_\phi$, $V_\theta$, $B \leftarrow \varnothing$
    **while** $\pi_\phi$ has not converged **do**
        **if** extend replay buffer $B$ **then**
            Extend $B$ using $\pi_\phi$ in $e$
        Sample trajectories from $B$
        Update $V_\theta$ using TD-error
        Update policy by following $\nabla_\phi L_\alpha$

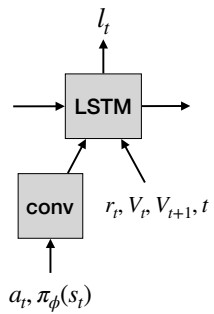

Figure 2: An overview of $L_\alpha(\tau, x(\phi), V)$.

### 3.3 GENERALITY AND EFFICIENCY OF METAGENRL

MetaGenRL offers a general framework for meta-learning objective functions that can represent a wide range of learning algorithms. In particular, it is only required that both $\pi_\phi$ and $L_\alpha$ can be differentiated w.r.t. to the policy parameters $\phi$. In the present work, we use this flexibility to leverage population-based meta-optimization, increase sample efficiency through off-policy second-order gradients, and to improve the generalization capabilities of meta-learned objective functions.

**Population-Based** A general objective function should be applicable to a wide range of environments and agent parameters. To this extent MetaGenRL is able to leverage the collective experience of *multiple* agents to perform meta-learning by using a *single* objective function $L_\alpha$ shared among a population of agents that each act in their own (potentially different) environment. Each agent locally computes Equation 6 over a batch of trajectories, and the resulting gradients are combined to update $L_\alpha$. Thus, the relevant learning experience of each individual agent is compressed into the objective function that is available to the entire population at any given time.

**Sample Efficiency** An alternative to learning neural objective functions using a population of agents is through evolution as in EPG (Houthooft et al., 2018). However, we expect meta-learning using second-order gradients as in MetaGenRL to be much more sample efficient. This is due to off-policy training of the objective function $L_\alpha$ and its subsequent off-policy use to improve the policy. Indeed, unlike in evolution there is no need to train multiple randomly initialized agents in their entirety in order to evaluate the objective function, thus speeding up credit assignment. Rather, at any point in time, any information that is deemed useful for future environment interactions can directly be incorporated into the objective function. Finally, using the formulation in Equation 6 one can measure the effects of improving the policy using $L_\alpha$ for *multiple* steps by increasing the corresponding number of gradient steps before applying $Q_\theta$, which we will explore in Section 5.2.3.

**Meta-Generalization** The focus of this work is to learn *general* learning rules that during test-time can be applied to vastly different environments. A strict separation between the policy and the learning rule, the functional form of the latter, and training across many environments all contribute to this. Regarding the former, a clear separation between the policy and the learning rule as in MetaGenRL is expected to be advantageous for two reasons. Firstly, it allows us to specify the number of parameters of the learning rule independent of the policy and critic parameters. For example, our implementation of $L_\alpha$ uses only $15K$ parameters for the objective function compared to $384K$ parameters for the policy and critic. Hence, we are able to only use a short description length for the learning rule. A second advantage that is gained is that the meta-learner is unable to directly change the policy and must, therefore, learn to make use of the objective function. This makes it difficult for the meta-learner to *overfit* to the training environments.

## 4 RELATED WORK

Among the earliest pursuits in meta-learning are meta-hierarchies of genetic algorithms (Schmidhuber, 1987) and learning update rules in supervised learning (Bengio et al., 1990). While the former introduced a general framework of entire meta-hierarchies, it relied on discrete non-differentiable programs. The latter introduced local update rules that included free parameters, which could be

learned using gradients in a supervised setting. Schmidhuber (1993) introduced a differentiable self-referential RNN that could address and modify its own weights, albeit difficult to learn.

Hochreiter et al. (2001) introduced differentiable meta-learning using RNNs to scale to larger problem instances. By giving an RNN access to its prediction error, it could implement its own meta-learning algorithm, where the weights are the meta-learned parameters, and the hidden states the subject of learning. This was later extended to the RL setting (Wang et al., 2016; Duan et al., 2016; Santoro et al., 2016; Mishra et al., 2018) (here refered to as RL$^2$). As we show empirically in our paper, meta-learning with RL$^2$ does not generalize well. It lacks a clear separation between policy and objective function, which makes it easy to overfit on training environments. This is exacerbated by the imbalance of $O(n^2)$ meta-learned parameters to learn $O(n)$ activations, unlike in MetaGenRL.

Many other recent meta-learning algorithms learn a policy parameter initialization that is later fine-tuned using a fixed reinforcement learning algorithm (Finn et al., 2017; Schulman et al., 2017; Grant et al., 2018; Yoon et al., 2018). Different from MetaGenRL, these approaches use second order gradients on the same policy parameter vector instead of using a separate objective function. Albeit in principle general (Finn & Levine, 2018), the mixing of policy and learning algorithm leads to a complicated way of expressing general update rules. Similar to RL$^2$, adaptation to related tasks is possible, but generalization is difficult (Houthooft et al., 2018).

Objective functions have been learned prior to MetaGenRL. Houthooft et al. (2018) evolve an objective function that is later used to train an agent. Unlike MetaGenRL, this approach is extremely costly in terms of the number of environment interactions required to evaluate and update the objective function. Most recently, Bechtle et al. (2019) introduced learned loss functions for reinforcement learning that also make use of second-order gradients, but use a policy gradient estimator instead of a Q-function. Similar to other work, their focus is only on narrow task distributions. Learned objective functions have also been used for learning unsupervised representations (Metz et al., 2019), DDPG-like meta-gradients for hyperparameter search (Xu et al., 2018), and learning from human demonstrations (Yu et al., 2018). Concurrent to our work, Alet et al. (2020) uses techniques from architecture search to search for viable artificial curiosity objectives that are composed of primitive objective functions.

Li & Malik (2016; 2017) and Andrychowicz et al. (2016) conduct meta-learning by learning optimizers that update parameters $\phi$ by modulating the gradient of some fixed objective function $L$: $\Delta\phi = f_\alpha(\nabla_\phi L)$ where $\alpha$ is learned. They differ from MetaGenRL in that they only modulate the gradient of a fixed objective function $L$ instead of learning $L$ itself.

Another connection exists to meta-learned intrinsic reward functions (Schmidhuber, 1991a; Dayan & Hinton, 1993; Wiering & Schmidhuber, 1996; Singh et al., 2004; Niekum et al., 2011; Zheng et al., 2018; Jaderberg et al., 2019). Choosing $\nabla_\phi L_\alpha = \tilde{\nabla}_\phi \sum_{t=1}^{T} \bar{r}_t(\tau)$, where $\bar{r}_t$ is a meta-learned reward and $\tilde{\nabla}_\theta$ is a gradient estimator (such as a value based or policy gradient based estimator) reveals that meta-learning objective functions includes meta-learning the gradient estimatior $\tilde{\nabla}$ itself as long as it is expressible by a gradient $\nabla_\theta$ on an objective $L_\alpha$. In contrast, for intrinsic reward functions, the gradient estimator $\tilde{\nabla}$ is normally fixed.

Finally, we note that positive transfer between different tasks (reward functions) as well as environments (e.g. different Atari games) has been shown previously in the context of transfer learning (Kistler et al., 1997; Parisotto et al., 2015; Rusu et al., 2016; 2019; Nichol et al., 2018) and meta-critic learning across tasks (Sung et al., 2017). In contrast to this work, the approaches that have shown to be successful in this domain rely entirely on human-engineered learning algorithms.

## 5 EXPERIMENTS

We investigate the learning and generalization capabilities of MetaGenRL on several continuous control benchmarks including *HalfCheetah (Cheetah)* and *Hopper* from MuJoCo (Todorov et al., 2012), and *LunarLanderContinuous (Lunar)* from OpenAI gym (Brockman et al., 2016). These environments differ significantly in terms of the properties of the underlying system that is to be controlled, and in terms of the dynamics that have to be learned to complete the environment. Hence, by training meta-RL algorithms on one environment and testing on other environments they provide a reasonable measure of out-of-distribution generalization.

Table 1: Mean return across multiple seeds (MetaGenRL: 6 meta-train $\times$ 2 meta-test seeds, RL$^2$: 6 meta-train $\times$ 2 meta-test seeds, EPG: 3 meta-train $\times$ 2 meta-test seeds) obtained by training randomly initialized agents during meta-test time on previously seen environments (cyan) and on unseen environments (brown). Boldface highlights best meta-learned algorithm. Mean returns (6 seeds) of several human-engineered algorithms are also listed.

| Training \ Testing | | Cheetah | Hopper | Lunar |
|---|---|---|---|---|
| **Cheetah & Hopper** | **MetaGenRL** | 2185 | **2439** | **18** |
| | **EPG** | -571 | 20 | -540 |
| | **RL$^2$** | **5180** | 289 | -479 |
| **Lunar & Cheetah** | **MetaGenRL** | **2552** | **2363** | 258 |
| | **EPG** | -701 | 8 | -707 |
| | **RL$^2$** | 2218 | 5 | **283** |
| **Lunar & Hopper & Walker & Ant** | **MetaGenRL (40 agents)** | 3106 | 2869 | 201 |
| **Cheetah & Lunar & Walker & Ant** | | 3331 | 2452 | -71 |
| **Cheetah & Hopper & Walker & Ant** | | 2541 | 2345 | -148 |
| | **PPO** | 1455 | 1894 | 187 |
| | **DDPG / TD3** | 8315 | 2718 | 288 |
| | **off-policy REINFORCE (GAE)** | -88 | 1804 | 168 |
| | **on-policy REINFORCE (GAE)** | 38 | 565 | 120 |

In our experiments, we will mainly compare to EPG and to RL$^2$ to evaluate the efficacy of our approach. We will also compare to several fixed model-free RL algorithms to measure how well the algorithms meta-learned by MetaGenRL compare to these handcrafted alternatives. Unless otherwise mentioned, we will meta-train MetaGenRL using 20 agents that are distributed equally over the indicated training environments[5]. Meta-learning uses clipped double-Q learning, delayed policy & objective updates, and target policy smoothing from TD3 (Fujimoto et al., 2018). We will allow for $600K$ environment interactions per agent during meta-training and then meta-test the objective function for $1M$ interactions. Further details are available in Appendix B.

## 5.1 COMPARISON TO PRIOR WORK

**Evaluating on previously seen environments**   We meta-train MetaGenRL on *Lunar* and compare its ability to train a randomly initialized agent at test-time (i.e. using the learned objective function and keeping it fixed) to DDPG, PPO, and on- and off-policy REINFORCE (both using GAE) across multiple seeds. Figure 3a shows that MetaGenRL markedly outperforms both the REINFORCE baselines and PPO. Compared to DDPG, which finds the optimal policy, MetaGenRL performs only slightly worse on average although the presence of outliers increases its variance. In particular, we find that some meta-test agents get 'stuck' for some time before reaching the optimal policy (see Section A.2 for additional analysis). Indeed, when evaluating only the best meta-learned objective function that was obtained during meta-training (MetaGenRL (best objective func) in Figure 3a) we are able to observe a strong reduction in variance and even better performance.

We also report results (Figure 3a) when meta-training MetaGenRL on both *Lunar* and *Cheetah*, and compare to EPG and RL$^2$ that were meta-trained on these same environments[6]. For MetaGenRL we were able to obtain similar performance to meta-training on only *Lunar* in this case. In contrast, for EPG it can be observed that even one billion environment interactions is insufficient to find a good objective function (in Figure 3a quickly dropping below -300). Finally, we find that RL$^2$ reaches the optimal policy after 100 million meta-training iterations, and that its performance is unaffected by additional steps during testing on *Lunar*. We note that RL$^2$ does not separate the policy and the learning rule and indeed in a similar 'within distribution' evaluation, RL$^2$ was found successful (Wang et al., 2016; Duan et al., 2016).

---

[5]An ablation study in Section A.3 revealed that a large number of agents is indeed required.

[6]In order to ensure a good baseline we allowed for a maximum of $100M$ environment interactions for RL$^2$ and $1B$ for EPG, which is more than eight / eighty times the amount used by MetaGenRL. Regarding EPG, this did require us to reduce the total number of seeds to 3 meta-train $\times$ 2 meta-test seeds.

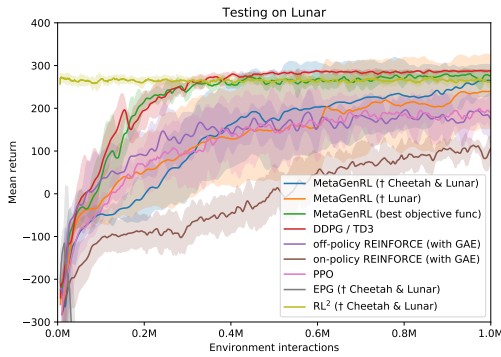 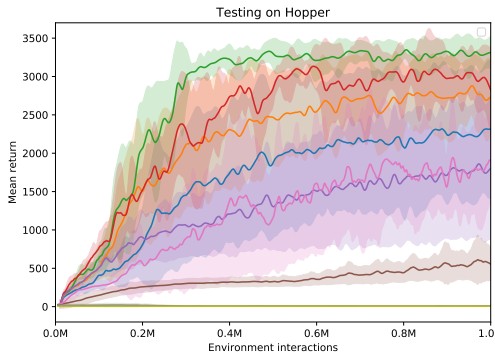

(a) Previously seen *Lunar* environment.      (b) Unseen *Hopper* environment.

Figure 3: Comparing the test-time training behavior of the meta-learned objective functions by MetaGenRL to other (meta) reinforcement learning algorithms. We train randomly initialized agents on (a) environments that were encountered during training, and (b) on significantly different environments that were unseen. Training environments are denoted by † in the legend. All runs are shown with mean and standard deviation computed over multiple random seeds (MetaGenRL: 6 meta-train $\times$ 2 meta-test seeds, $RL^2$: 6 meta-train $\times$ 2 meta-test seeds, EPG: 3 meta-train $\times$ 2 meta-test seeds, and 6 seeds for all others).

Table 1 provides a similar comparison for two other environments. Here we find that in general MetaGenRL is able to outperform the REINFORCE baselines and PPO, and in most cases (except for *Cheetah*) performs similar to DDPG[7]. We also find that MetaGenRL consistently outperforms EPG, and often $RL^2$. For an analysis of meta-training on more than two environments we refer to Appendix A.

**Generalization to vastly different environments**  We evaluate the same objective functions learned by MetaGenRL, EPG and the recurrent dynamics by $RL^2$ on *Hopper*, which is significantly different compared to the meta-training environments. Figure 3b shows that the learned objective function by MetaGenRL continues to outperform both PPO and our implementations of REINFORCE, while the best performing configuration is even able to outperform DDPG.

When comparing to related meta-RL approaches, we find that MetaGenRL is significantly better in this case. The performance of EPG remains poor, which was expected given what was observed on previously seen environments. On the other hand, we now find that the $RL^2$ baseline fails completely (resulting in a flat low-reward evaluation), suggesting that the learned learning rule that was previously found to be successful is in fact entirely overfitted to the environments that were seen during meta-training. We were able to observe similar results when using different train and test environment splits as reported in Table 1, and in Appendix A.

## 5.2 ANALYSIS

### 5.2.1 META-TRAINING PROGRESSION OF OBJECTIVE FUNCTIONS

Previously we focused on test-time training randomly initialized agents using an objective function that was meta-trained for a total of $600K$ steps (corresponding to a total of $12M$ environment interactions across the entire population). We will now investigate the quality of the objective functions during meta-training.

Figure 4 displays the result of evaluating an objective function on *Hopper* at different intervals during meta-training on *Cheetah* and *Lunar*. Initially ($28K$ steps) it can be seen that due to lack of meta-training there is only a marginal improvement in the return obtained during test time. However, after only meta-training for $86K$ steps we find (perhaps surprisingly) that the meta-trained

---

[7]We emphasize that the neural objective function under consideration is unable to implement DDPG and only uses a constant value estimate (i.e. $\nabla_\phi V = 0$ by using gradient stopping) during meta testing.

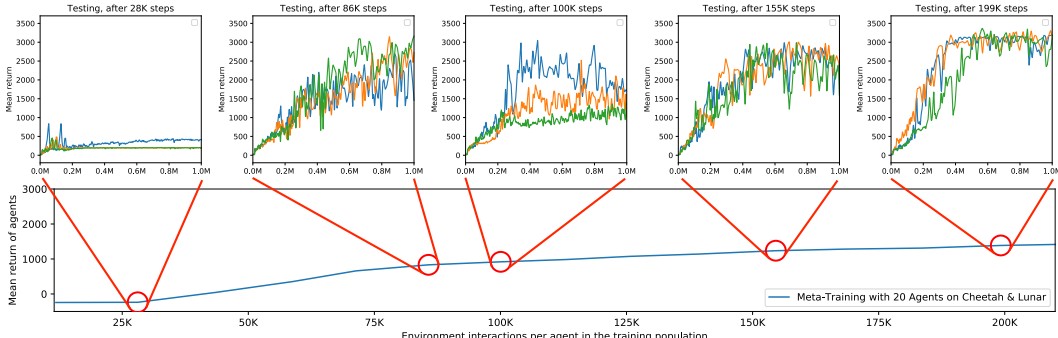

Figure 4: Meta-training with 20 agents on *Cheetah* and *Lunar*. We test the objective function at five stages of meta-training by using it to train three randomly initialized agents on *Hopper*.

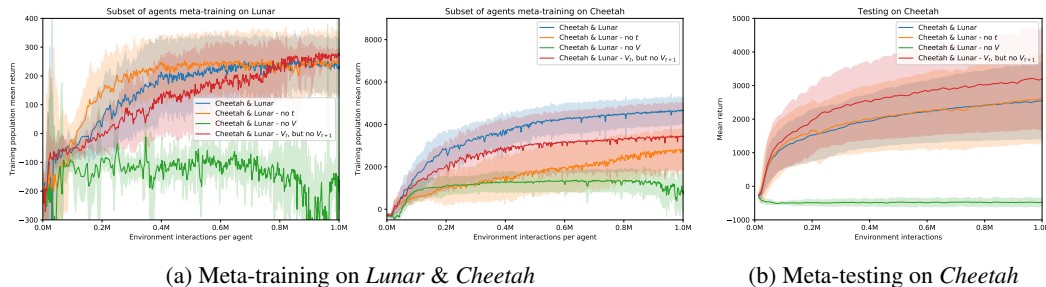

(a) Meta-training on *Lunar* & *Cheetah*                    (b) Meta-testing on *Cheetah*

Figure 5: We meta-train MetaGenRL using several alternative parametrizations of $L_\alpha$ on a) *Lunar* and *Cheetah*, and b) present results of testing on *Cheetah*. During meta-training a representative example of a single agent population is shown with shaded regions denoting standard deviation across the population. Meta-test results are reported as per usual across 6 meta-train $\times$ 2 meta-test seeds.

objective function is already able to make consistent progress in optimizing a randomly initialized agent during test-time. On the other hand, we observe large variances at test-time during this phase of meta-training. Throughout the remaining stages of meta-training we then observe an increase in convergence speed, more stable updates, and a lower variance across seeds.

### 5.2.2 ABLATION STUDY

We conduct an ablation study of the neural objective function that was described in Section 3.2. In particular, we assess the dependence of $L_\alpha$ on the value estimates $V_t, V_{t+1}$ and on the time component that could to some extent be learned. Other ablations, including limiting access to the action chosen or to the received reward, are expected to be disastrous for generalization to any other environment (or reward function) and therefore not explored.

**Dependence on $t$**    We use a parameterized objective function of the form $L_\alpha(a_t, r_t, V_t, \pi_\phi(s_t)|t \in 0, ..., T-1)$ as in Figure 2 except that it does not receive information about the time-step $t$ at each step. Although information about the current time-step is required in order to learn (for example) a generalized advantage estimate (Schulman et al., 2015b), the LSTM could in principle learn such time tracking on it own, and we expect only minor effects on meta-training and during meta-testing. Indeed in Figure 5b it can be seen that the neural objective function performs well without access to $t$, although it converges slower on *Cheetah* during meta-training (Figure 5a).

**Dependence on $V$**    We use a parameterized objective function of the form $L_\alpha(a_t, r_t, t, \pi_\phi(s_t)|t \in 0, ..., T-1)$ as in Figure 2 except that it does not receive any information about the value estimates at time-step $t$. There exist reinforcement learning algorithms that work without value function estimates (eg. Williams (1992); Schmidhuber & Zhao (1998)), although in the absence of an alternative baseline these often have a large variance. Similar results are observed for this ablation in Figure 5a

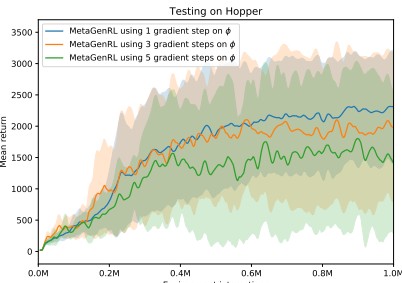 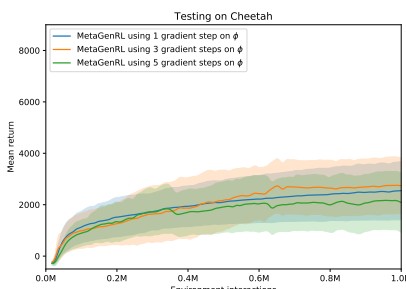

Figure 6: We meta-train MetaGenRL on the *LunarLander* and *HalfCheetah* environments using one, three, and five inner gradient steps on $\phi$. Meta-test results are reported across 3 meta-train $\times$ 2 meta-test seeds.

during meta-training where a possibly large variance appears to affect meta-training. Correspondingly during test-time (Figure 5b) we do not find any meaningful training progress to take place. In contrast, we find that we can remove the dependence on *one* of the value function estimates, i.e. remove $V_{t+1}$ but keep $V_t$, which during some runs even increases performance.

### 5.2.3 MULTIPLE GRADIENT STEPS

We analyze the effect of making *multiple* gradient updates to the policy using $L_\alpha$ before applying the critic to compute second-order gradients with respect to the objective function parameters as in Equation 6. While in previous experiments we have only considered applying a single update, multiple gradient updates might better capture long term effects of the objective function. At the same time, moving further away from the current policy parameters could reduce the overall quality of the second-order gradients. Indeed, in Figure 6 it can be observed that using 3 gradient steps already slightly increases the variance during test-time training on *Hopper* and *Cheetah* after meta-training on *LunarLander* and *Cheetah*. Similarly, we find that further increasing the number of gradient steps to 5 harms performance.

## 6 CONCLUSION

We have presented MetaGenRL, a novel off-policy gradient-based meta reinforcement learning algorithm that leverages a population of DDPG-like agents to meta-learn general objective functions. Unlike related methods the meta-learned objective functions do not only generalize in narrow task distributions but show similar performance on entirely different tasks while markedly outperforming REINFORCE and PPO. We have argued that this generality is due to MetaGenRL's explicit separation of the policy and learning rule, the functional form of the latter, and training across multiple agents and environments. Furthermore, the use of second order gradients increases MetaGenRL's sample efficiency by several orders of magnitude compared to EPG (Houthooft et al., 2018).

In future work, we aim to further improve the learning capabilities of the meta-learned objective functions, including better leveraging knowledge from prior experiences. Indeed, in our current implementation, the objective function is unable to observe the environment or the hidden state of the (recurrent) policy. These extensions are especially interesting as they may allow more complicated curiosity-based (Schmidhuber, 1991b; 1990; Houthooft et al., 2016; Pathak et al., 2017) or model-based (Schmidhuber, 1990; Weber et al., 2017; Ha & Schmidhuber, 2018) algorithms to be learned. To this extent, it will be important to develop introspection methods that analyze the learned objective function and to scale MetaGenRL to make use of many more environments and agents.

ACKNOWLEDGEMENTS

We thank Paulo Rauber, Imanol Schlag, and the anonymous reviewers for their feedback. This work was supported by the ERC Advanced Grant (no: 742870) and computational resources by the Swiss National Supercomputing Centre (CSCS, project: s978). We also thank NVIDIA Corporation for donating a DGX-1 as part of the Pioneers of AI Research Award and to IBM for donating a Minsky machine.

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

# A    ADDITIONAL RESULTS

## A.1    ALL TRAINING AND TEST REGIMES

In the main text, we have shown several combinations of meta-training, and testing environments. We will now show results for all combinations, including the respective human engineered baselines.

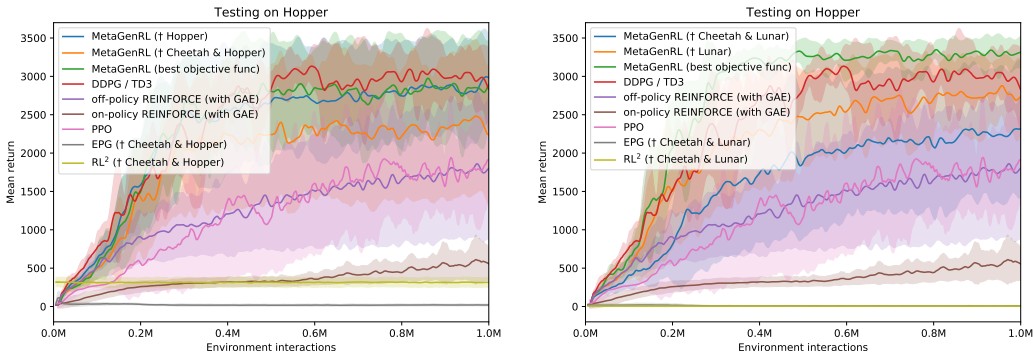

(a) Within distribution generalization.                (b) Out of distribution generalization.

Figure 7: Comparing the test-time training behavior of the meta-learned objective functions by MetaGenRL to other (meta) reinforcement learning algorithms on *Hopper*. We consider within distribution testing (a), and out of distribution testing (b) by varying the meta-training environments (denoted by †) for the meta-RL approaches. All runs are shown with mean and standard deviation computed over multiple random seeds (MetaGenRL: 6 meta-train × 2 meta-test seeds, RL$^2$: 6 meta-train × 2 meta-test seeds, EPG: 3 meta-train × 2 meta-test seeds, and 6 seeds for all others).

**Hopper**    On *Hopper* (Figure 7) we find that MetaGenRL works well, both in terms of generalization to previously seen environments, and to unseen environments. The PPO, REINFORCE, RL$^2$, and EPG baselines are outperformed significantly. Regarding RL$^2$ we observe that it is only able to obtain reward when *Hopper* was included during meta-training, although its performance is generally poor. Regarding EPG, we observe some learning progress during meta-testing on *Hopper* after meta-training on *Cheetah* and *Hopper* (Figure 7a), although it drops back down quickly as test-time training proceeds. In contrast, when meta-testing on *Hopper* after meta-training on *Cheetah* and *Lunar* (Figure 7b) no test-time training progress is observed at all.

**Cheetah**    Similar results are observed in Figure 8 for *Cheetah*, where MetaGenRL outperforms PPO and REINFORCE significantly. On the other hand, it can be seen that DDPG notably outperforms MetaGenRL on this environment. It will be interesting to further study these differences in the future to improve the expressibility of our approach. Regarding RL$^2$ and EPG only within distribution generalization results are available due to *Cheetah* having larger observations and / or action spaces compared to *Hopper* and *Lunar*. We observe that RL$^2$ performs similar to our earlier findings on *Hopper* but significantly improves in terms of within-distribution generalization (likely due to greater overfitting, as was consistently observed for other splits). EPG shows initially more promise on within distribution generalization (Figure 8a), but ends up like before.

**Lunar**    On *Lunar* (Figure 9) we find that MetaGenRL is only marginally better compared to the REINFORCE and PPO baselines in terms of within distribution generalization and worse in terms of out of distribution generalization. Analyzing this result reveals that although many of the runs train rather well, some get stuck during the early stages of training without or only delayed recovering. These outliers lead to a seemingly very large variance for MetaGenRL in Figure 9b. We will provide a more detailed analysis of this result in Section A.2. If we focus on the best performing objective function then we observe competitive performance to DDPG (Figure 9a). Nonetheless, we notice that the objective function trained on *Hopper* generalizes worse to *Lunar*, despite our earlier result that objective functions trained on *Lunar* do in fact generalize well to *Hopper*. MetaGenRL is still able to outperform both RL$^2$ and EPG in terms of out of distribution generalization. We do note

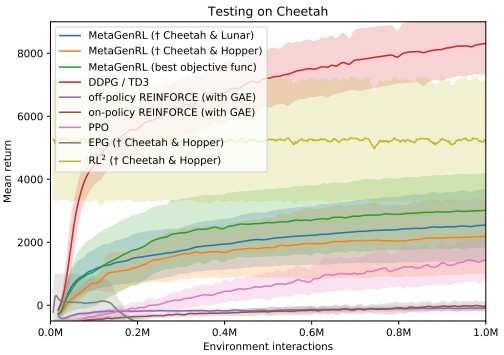 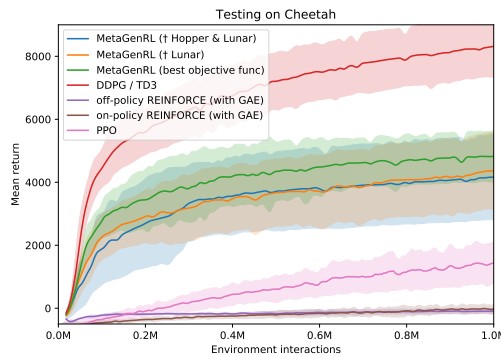

(a) Within distribution generalization.
(b) Out of distribution generalization.

Figure 8: Comparing the test-time training behavior of the meta-learned objective functions by MetaGenRL to other (meta) reinforcement learning algorithms on *Cheetah*. We consider within distribution testing (a), and out of distribution testing (b) by varying the meta-training environments (denoted by †) for the meta-RL approaches. All runs are shown with mean and standard deviation computed over multiple random seeds (MetaGenRL: 6 meta-train × 2 meta-test seeds, RL$^2$: 6 meta-train × 2 meta-test seeds, EPG: 3 meta-train × 2 meta-test seeds, and 6 seeds for all others).

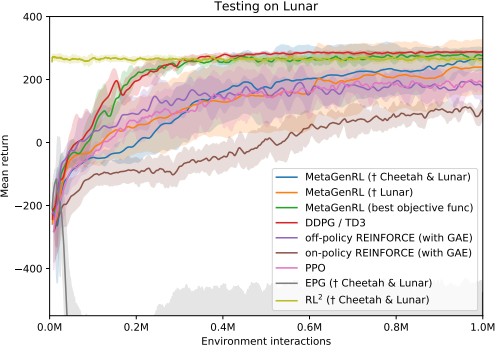 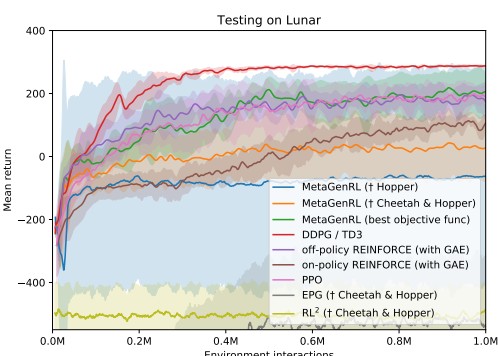

(a) Within distribution generalization.
(b) Out of distribution generalization.

Figure 9: Comparing the test-time training behavior of the meta-learned objective functions by MetaGenRL to other (meta) reinforcement learning algorithms on *Lunar*. We consider within distribution testing (a), and out of distribution testing (b) by varying the meta-training environments (denoted by †) for the meta-RL approaches. All runs are shown with mean and standard deviation computed over multiple random seeds (MetaGenRL: 6 meta-train × 2 meta-test seeds, RL$^2$: 6 meta-train × 2 meta-test seeds, EPG: 3 meta-train × 2 meta-test seeds, and 6 seeds for all others).

that EPG is able to meta-learn objective functions that are able to improve to some extent during test time.

**Comparing final scores**  An overview of the final scores that were obtained for MetaGenRL in comparison to the human engineered baselines is shown in Table 2. It can be seen that MetaGenRL outperforms PPO and off-/on-policy REINFORCE in most configurations while DDPG with TD3 tricks remains stronger on two of the three environments. Note that DDPG is currently not among the representable algorithms by MetaGenRL.

## A.2 STABILITY OF LEARNED OBJECTIVE FUNCTIONS

In the results presented in Figure 9 on *Lunar* we observed a seemingly large variance for MetaGenRL that was due to outliers. Indeed, when analyzing the individual runs meta-trained on *Lunar* and tested on *Lunar* we found that that one of the runs converged to a local optimum early on during

Table 2: Agent mean return across multiple seeds (MetaGenRL: 6 meta-train $\times$ 2 meta-test seeds, and 6 seeds for all others) for meta-test training on previously seen environments (cyan) and on unseen (different) environments (brown) compared to human engineered baselines.

| | **Training (below) / Test (right)** | Cheetah | Hopper | Lunar |
|---|---|---|---|---|
| **MetaGenRL (20 agents)** | **Cheetah & Hopper** | 2185 | 2433 | 18 |
| | **Cheetah & Lunar** | 2551 | 2363 | 258 |
| | **Hopper & Lunar** | 4160 | **2966** | 146 |
| | **Hopper** | 3646 | 2937 | -62 |
| | **Lunar** | 4366 | 2717 | 244 |
| **MetaGenRL (40 agents)** | **Lunar & Hopper & Walker & Ant** | 3106 | 2869 | 201 |
| | **Cheetah & Lunar & Walker & Ant** | 3331 | 2452 | -71 |
| | **Cheetah & Hopper & Walker & Ant** | 2541 | 2345 | -148 |
| **PPO** | - | 1455 | 1894 | 187 |
| **DDPG / TD3** | - | **8315** | 2718 | **288** |
| **off-policy REINFORCE (GAE)** | - | -88 | 1804 | 168 |
| **on-policy REINFORCE (GAE)** | - | 38 | 565 | 120 |

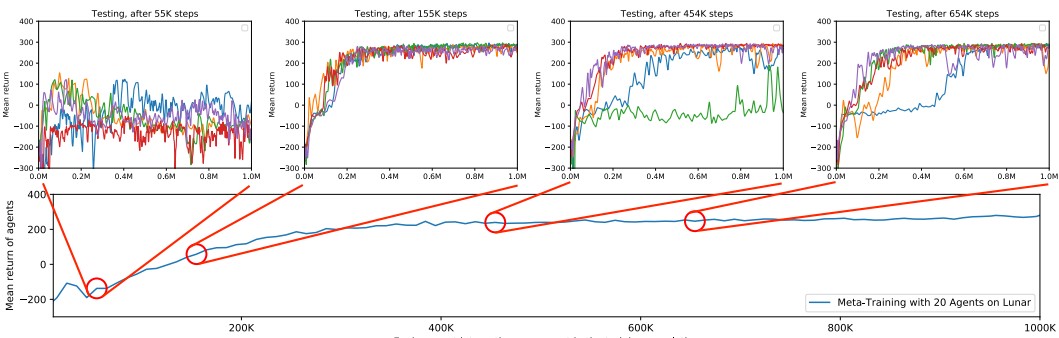

Figure 10: Meta-training with 20 agents on *LunarLander*. We meta-test the objective function at different stages in training on the same environment.

training and was unable to recover from this afterwards. On the other hand, we also observed that runs can be 'stuck' for a long time to then make very fast learning progress. It suggests that the objective function may sometimes experience difficulties in providing meaningful updates to the policy parameters during the early stages of training.

We have further analyzed this issue by evaluating one of the objective functions at several intervals throughout meta-training in Figure 10. From the meta-training curve (bottom) it can be seen that meta-training in *Lunar* converges very early. This means that from then on, updates to the objective function will be based on mostly converged policies. As the test-time plots show, these additional updates appear to negatively affect test-time performance. We hypothesize that the objective function essentially 'forgets' about the early stages of training a randomly initialized agent, by only incorporating information about good performing agents. A possible solution to this problem would be to keep older policies in the meta-training agent population or use early stopping.

Finally, if we exclude four random seeds (of 12), we indeed find a significant reduction in the variance (and increase in the mean) of the results observed for MetaGenRL (see Figure 11).

### A.3 ABLATION OF AGENT POPULATION SIZE AND UNIQUE ENVIRONMENTS

In our experiments we have used a population of 20 agents during meta-training to ensure diversity in the conditions under which the objective function needs to optimize. The size of this population is a crucial parameter for a stable meta-optimization. Indeed, in Figure 12 it can be seen that meta-training becomes increasingly unstable as the number of agents in the population decreases.

Using a similar argument, one would expect to gain from increasing the number of distinct environments (or agents) during meta-training. In order to verify this, we have evaluated two additional

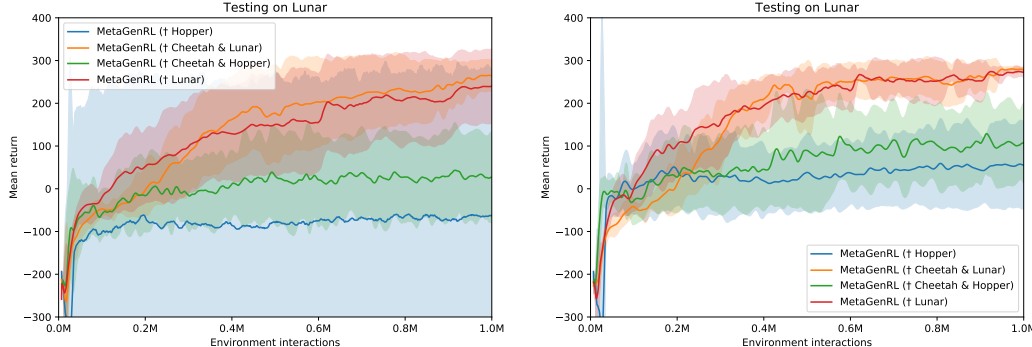

Figure 11: The left plot shows all 12 random seeds on the meta-test environment *Lunar* while the right has the 4 worst random seeds removed. The variance is now reduced significantly.

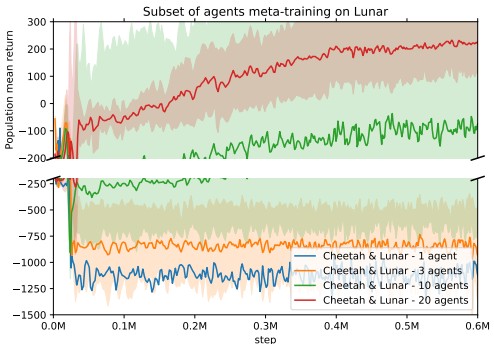

Figure 12: Stable meta-training requires a large population size of at least 20 agents. Meta-training performance is shown for a single run with the mean and standard deviation across the agent population.

Figure 13: Meta-training on *Cheetah*, *Lunar*, *Walker*, and *Ant* with 20 or 40 agents; meta-testing on the out-of-distribution *Hopper* environment. We compare to previous MetaGenRL configurations.

settings: Meta-training on *Cheetah & Lunar & Walker & Ant* with 20 and 40 agents respectively. Figure 13 shows the result of meta-testing on *Hopper* for these experiments (also see the final results reported for 40 agents in Table 2). Unexpectedly, we find that increasing the number of distinct environments does not yield a significant improvement and, in fact, sometimes even decrease performance. One possibility is that this is due to the simple form of the objective function under consideration, which has no access to the environment observations to efficiently distinguish between them. Another possibility is that MetaGenRL's hyperparameters require additional tuning in order to be compatible with these setups.

## B EXPERIMENT DETAILS

In the following we describe all experimental details regarding the architectures used, meta-training, hyperparameters, and baselines. The code to reproduce our experiments is available at `http://louiskirsch.com/code/metagenrl`.

### B.1 NEURAL OBJECTIVE FUNCTION ARCHITECTURE

**Neural Architecture**   In this work we use an LSTM to implement the objective function (Figure 2). The LSTM runs backwards in time over the state, action, and reward tuples that were encountered during the trajectory $\tau$ under consideration. At each step $t$ the LSTM receives as input the reward $r_t$, value estimates of the current and previous state $V_t, V_{t+1}$, the current timestep $t$ and finally the action that was taken at the current timestep $a_t$ in addition to the action as determined by the current policy $\pi_\phi(s_t)$. The actions are first processed by one dimensional convolutional layers striding over the action dimension followed by a reduction to the mean. This allows for different action sizes between environments. Let $A^{(B)} \in \mathbb{R}^{1 \times D}$ be the action from the replay buffer, $A^{(\pi)} \in \mathbb{R}^{1 \times D}$ be the action predicted by the policy, and $W \in \mathbb{R}^{2 \times N}$ a learnable matrix corresponding to $N$ outgoing units, then the actions are transformed by

$$\frac{1}{D} \sum_{i=1}^{D} ([A^{(B)}, A^{(\pi)}]^T W)_i, \tag{7}$$

where $[a, b]$ is a concatenation of $a$ and $b$ along the first axis. This corresponds to a convolution with kernel size 1 and stride 1. Further transformations with non-linearities can be added after applying $W$, if necessary. We found it helpful (but not strictly necessary) to use ReLU activations for half of the units and square activations for the other half.

At each time-step the LSTM outputs a scalar value $l_t$ (bounded between $-\eta$ and $\eta$ using a scaled tanh activation), which are summed to obtain the value of the neural objective function. Differentiating this value with respect to the policy parameters $\phi$ then yields gradients that can be used to improve $\pi_\phi$. We only allow gradients to flow backwards through $\pi_\phi(s_t)$ to $\phi$. This implementation is closely related to the functional form of a REINFORCE (Williams, 1992) estimator using the generalized advantage estimation (Schulman et al., 2015b).

All feed-forward networks (critic and policy) use ReLU activations and layer normalization (Ba et al., 2016). The LSTM uses tanh activations for cell and hidden state transformations, sigmoid activations for the gates. The input time $t$ is normalized between 0 at the beginning of the episode and 1 at the final transition. Any other hyper-parameters can be seen in Table 3.

**Extensibility**   The expressability of the objective function can be further increased through several means. One possibility is to add the entire sequence of state observations $o_{1:T}$ to its inputs, or by introducing a bi-directional LSTM. Secondly, additional information about the policy (such as the hidden state of a recurrent policy) can be provided to $L$. Although not explored in this work, this would in principle allow one to learn an objective that encourages certain representations to emerge, e.g. a predictive representation about future observations, akin to a world model (Schmidhuber, 1990; Ha & Schmidhuber, 2018; Weber et al., 2017). In turn, these could create pressure to adapt the policy's actions to explore unknown dynamics in the environment (Schmidhuber, 1991b; 1990; Houthooft et al., 2016; Pathak et al., 2017).

### B.2 META-TRAINING

**Annealing with DDPG**   At the beginning of meta-training (learning $L_\alpha$), the objective function is randomly initialized and thus does not make sensible updates to the policies. This can lead to irreversibly breaking the policies early during training. Our current implementation circumvents this issue by linearly annealing $\nabla_\phi L_\alpha$ the first 10k timesteps ($\sim 2\%$ of all timesteps) with DDPG $\nabla_\phi Q_\theta(s_t, \pi_\phi(s_t))$. Preliminary experiments suggested that an exponential learning rate schedule on the gradient of $\nabla_\phi L_\alpha$ for the first 10k steps can replace the annealing with DDPG. The learning rate anneals exponentially between a learning rate of zero and 1e-3. However, in some rare cases this may still lead to unsuccessful training runs, and thus we have omitted this approach from the present work.

**Standard training** During training, the critic is updated twice as many times as the policy and objective function, similar to TD3 (Fujimoto et al., 2018). One gradient update with data sampled from the replay buffer is applied for every timestep collected from the environment. The gradient with respect to $\phi$ in Equation 6 is combined with $\phi$ using a fixed learning rate in the standard way, all other parameter updates use Adam (Kingma & Ba, 2015) with the default parameters. Any other hyper-parameters can be seen in Table 3 and Table 4.

**Using additional gradient steps** In our experiments (Section 5.2.3) we analyzed the effect of applying *multiple* gradient updates to the policy using $L_\alpha$ before applying the critic to compute second-order gradients with respect to the objective function parameters. For two updates, this gives

$$\nabla_\alpha Q_\theta(s_t, \pi_{\phi^\dagger}(s_t)) \text{ with } \phi^\dagger = \phi' - \nabla_{\phi'} L_\alpha(\tau_1, x(\phi'), V)$$
$$\text{and } \phi' = \phi - \nabla_\phi L_\alpha(\tau_2, x(\phi), V) \tag{8}$$

and can be extended to more than two correspondingly. Additionally, we use disjoint mini batches of data $\tau$: $\tau_1, \tau_2$. When updating the policy using $\nabla_\phi L_\alpha$ we continue to use only a single gradient step.

### B.3 BASELINES

**RL$^2$** The implementation for RL$^2$ mimics the paper by Duan et al. (Duan et al., 2016). However, we were unable to achieve good results with TRPO (Schulman et al., 2015a) on the MuJoCo environments and thus used PPO (Schulman et al., 2017) instead. The PPO hyperparameters and implementation are taken from rllib (Liang et al., 2018). Our implementation uses an LSTM with 64 units and does not reset the state of the LSTM for two episodes in sequence. Resetting after additional episodes were given did not improve training results. Different action and observation dimensionalities across environments were handled by using an environment wrapper that pads both with zeros appropriately.

**EPG** We use the official EPG code base `https://github.com/openai/EPG` from the original paper (Houthooft et al., 2018). The hyperparameters are taken from the paper, $V = 64$ noise vectors, an update frequency of $M = 64$, and 128 updates for every inner loop, resulting in an inner loop length of 8196 steps. During meta-test training, we run with the same update frequency for a total of 1 million steps.

**PPO & On-Policy REINFORCE with GAE** We use the tuned implementations from `https://spinningup.openai.com/en/latest/spinningup/bench.html` which include a GAE (Schulman et al., 2015b) baseline.

**Off-Policy Reinforce with GAE** The implementation is equivalent to MetaGenRL except that the objective function is fixed to be the REINFORCE estimator with a GAE (Schulman et al., 2015b) baseline. Thus, experience is sampled from a replay buffer. We have also experimented with an importance weighted unbiased estimator but this resulted in poor performance.

**DDPG** Our implementation is based on `https://spinningup.openai.com/en/latest/spinningup/bench.html` and uses the same TD3 tricks (Fujimoto et al., 2018) and hyperparameters (where applicable) that MetaGenRL uses.

Table 3: Architecture hyperparameters

| Parameter | Value |
| --- | --- |
| Critic number of layers | 3 |
| Critic number of units | 350 |
| Policy number of layers | 3 |
| Policy number of units | 350 |
| Objective function LSTM units | 32 |
| Objective function action conv layers | 3 |
| Objective function action conv filters | 32 |
| Error bound $\eta$ | 1000 |

Table 4: Training hyperparameters

| Parameter | Value |
| --- | --- |
| Truncated episode length | 20 |
| Global norm gradient clipping | 1.0 |
| Critic learning rate $\lambda_1$ | 1e-3 |
| Policy learning rate $\lambda_2$ | 1e-3 |
| Second order learning rate $\lambda_3$ | 1e-3 |
| Obj. func. learning rate $\lambda_4$ | 1e-3 |
| Critic noise | 0.2 |
| Critic noise clip | 0.5 |
| Target network update speed | 0.005 |
| Discount factor | 0.99 |
| Batch size | 100 |
| Random exploration timesteps | 10000 |
| Policy gaussian noise std | 0.1 |
| Timesteps per agent | 1M |

