# OpenReview forum: "Improving Generalization in Meta Reinforcement Learning using Learned Objectives"
_ICLR.cc/2020/Conference — Accept (Spotlight)_

### Official Review · AnonReviewer1 · 2019-10-20
**Official Blind Review #1**

**Rating:** 8

**Review:**

The paper proposes a meta reinforcement learning algorithm called MetaGenRL, which meta-learns learning rules to generalize to different environments. The paper poses an important observation where learning rules in reinforcement learning to train the agents are results of human engineering and design, instead, the paper demonstrates how to use second-order gradients to learn learning rules to train agents. Learning learning rules in general has been proposed and this paper is another attempt to further generalize what could be learned in the learning rules. The idea is verified on three Mujoco domains, where the neural objective function is learned from one / two domains, then deployed to a new unseen domain. The experiments show that the learned neural objective can generalize to new environments which are different from the meta-training environments.

Overall, the paper is a novel paper and with clear motivation, I like the paper a lot! Hope that the authors could address the following concerns and make the paper even better:

1. The current experiment setup is a great proof-of-concept, however it seems a bit limited to support the claims in the paper. The meta-training has only at most two environments and the generalization of the neural objective function is only performed at one environment. It would be great if the authors could show more results with more meta-training environments (say, 10 meta-training environments) and more meta-testing environments (the current setup is only with one);

2. The paper states a hypothesis that LSTM as a general function approximator, it is in principle able to learn variance and bias reduction techniques. However, in practice, due to learning dynamics and many other factors, it's not necessary true, i.e., how many samples are required for an LSTM to learn such technique is unclear. At the same time, at Page 8, Section "Dependence on V" actually acts as an example of LSTM couldn't figure out an effective variance-reduction method during the short meta-training time. The authors may want to put more words around the learnability of variance-bias trade-off techniques.

Notation issues which could be further improved:
1.  Page 2, "Notation" section and all of the following time indexing. Note that in Equation (1), r(s_1, a_t) has discount gamma^1, which is not true, I'd recommend the authors to follow the time indexing starting from 0, so that the Equation (1) is correct. (Alternatively, the authors could change from gamma^t into gamma^{t-1});
2. Section "Human Engineered Gradient Estimators" is missing the formal introduction of the notation \tau;
3. Overall, the authors seem to use \Phi and \theta interchangeably, it's better to use a unified notation across the paper;
4. In the paper, the authors choose \alpha to represent the neural net for learning the objective function, to make it clearer for the readers, the authors could consider to change \alpha into \eta, because \alpha is often considered as learning rate notation;
5. I'd suggest the authors to rewrite the paragraph in Page 3 "MetaGrenRL builds on this idea of ....,  using L_\alpha on the estimated return". This describes a key step in the algorithm while at the moment it's not very clear to the readers what's going on there;
6. Section 3.1 is missing a step to go from Q into V;
7. The authors could consider to describe the details of the algorithms in a more general actor-critic form, instead of starting from DDPG formulation. It would make the methods more general applicable (for example, extension to discrete action space).



**Experience Assessment:**

I have published one or two papers in this area.

**Review Assessment: Checking Correctness Of Derivations And Theory:**

I carefully checked the derivations and theory.

**Review Assessment: Checking Correctness Of Experiments:**

I assessed the sensibility of the experiments.

**Review Assessment: Thoroughness In Paper Reading:**

I read the paper thoroughly.

---

> ### Author Response · Authors · 2019-11-09
> **Response to reviewer #1 with comments on improvements**
>
> Thank you for your review and valuable feedback!
>
> Below we will address each of your comments in detail.
>
> (1) We agree that it would be desirable to consider additional environments, including meta-training on more than 2 environments. However, using a large population of agents and many environments requires a lot of compute that was not available at the time of submission. Nonetheless, it is our goal to incorporate a single large experiment (meta-training on many environments using many agents) to demonstrate the capabilities of MetaGenRL in this regime and to further support our claims.
>
> Regarding additional meta-test environments, note that the current experiments already consider multiple meta-test environments. For example, table 1 shows two configurations of meta-training environments and three meta-test environments each.
>
> (2) We agree that it is unlikely that the RNN in practice will learn known variance and bias reduction techniques (although it would be hard to evaluate that). Indeed, based on the drop in performance when not providing V one could argue that the RNN was unable to learn an effective variance-reduction technique, although there could be other factors at play. For example, note that when V was included, our experiments showed that the objective function worked better in many tested cases than commonly used policy gradient algorithms such as REINFORCE with the Generalized Advantage (GAE) estimator, PPO with the GAE, or off-policy REINFORCE with GAE.
>
> Finally, we would like to thank you for pointing out various notational issues and suggestions for improving clarity. We have incorporated several of these to improve the paper.

---

### Official Review · AnonReviewer2 · 2019-10-23
**Official Blind Review #2**

**Rating:** 6

**Review:**

Summary:
This paper presents a novel meta reinforcement learning algorithm capable of meta-generalizing to unseen tasks. They make use of a learned objective function used in combination with DDPG style update. Results are presented on different combinations of meta-training and meta-testing on lunar, half cheetah, and hopper environments with a focus on meta-generalization to vastly different environments.
Motivation:
The work is well motivated and is tackling an important problem. There are a number of design decisions presented and only some are validated experimentally. Given the complexity of many existing meta-rl methods this seems fine but could obviously be improved upon either with more empirical work or with some guiding theory.
Experiments:
Overall the experiments are not convincing to me. Given that this is the majority of your paper is empirically based this is my main criticism. More detailed comments follow.
Figure 2a concerns me that just meta-training on lunar performs worse than ddpg (what your algorithm is based on). This suggests that this added complexity is not aiding in capacity and or hurting training. Can you comment on this? This result also casts doubt onto figure 2b, which, in isolation seems like an extremely promising example of meta-generalization. This makes me fear there is something indirect and not interesting occurring (e.g. the learned loss modifies with the DDPG algorithm which happens to increase noise in generated samples which improve performance only on some environments and hurts in others for example.)
Table 1 should include hand designed algorithms imo. Given how weak EPG is (as you stated for number of frames) and how RL^2 will never generalize across these different tasks it's hard to get a sense of the numbers. Your appendix does include a figure like this which shows ddpg performs quite well. Additionally, I don't understand why meta-training on lunar and transferring to hopper does better than meta-training on hopper (table 1, middle column). Can you comment on this?
While figure 3 is cool, I would appreciate if it put the meta-test performance on the same graph as meta-train performance. From eyeballing the curves it looks like it decreases at 100k iterations then finally increases again at 200k. This is strange. This also seems fraught from an empirical comparison point of view. How do you select when to test these algorithms? Ideally you would have a meta-validation set of tasks then only meta-test on the selected task but I see no mention of this.
Key details such as meta-training are also not discussed in depth nor ablated. From the details and curriculum scheme presented in the appendix this seems like quite a feat. Further study of these factors could be useful.
Hyperparameters of your baseline do not appear to be tuned (taken from appendix) where as for your method has a number of choices. How are you tuning these choices? Once again a meta-validation set would be the principled thing to tune against.
Finally, the experimental setup presented here is quite complicated. There are a ton of factors at play -- exploration, meta-generalization, meta-training, inner-training, instability of ddpg, so on. These all complicate the resulting picture. Having some simplified / more controlled setup to demonstrate these pieces would be greatly appreciated.
Other Suggestions:
Section 3 generalization: I think you mean meta-generalization.
Please include what error bars are for all plots.

Rating:
I am borderline leaning towards reject on this paper. I enjoyed reading this work and found the ideas interesting but the empirical comparisons are confusing and not convincing. I hope the authors continue to work to improve this!


**Experience Assessment:**

I have published one or two papers in this area.

**Review Assessment: Checking Correctness Of Derivations And Theory:**

N/A

**Review Assessment: Checking Correctness Of Experiments:**

I carefully checked the experiments.

**Review Assessment: Thoroughness In Paper Reading:**

I read the paper at least twice and used my best judgement in assessing the paper.

---

> ### Author Response · Authors · 2019-11-09
> **Response to reviewer #2 with comments on improvements [1/2]**
>
> Thank you for your review and valuable feedback!
>
> Before we proceed into detail we would first like to clarify some aspects that motivated this work, including our choice of experiments, and baselines.
>
> The premise of meta-RL is (1) that meta-learning learning rules for RL allows us to outperform existing human-engineered approaches on single environments, and (2) that learned learning rules are able to incorporate knowledge about learning in one task (or environment), to improve learning others. While prior work in meta-RL has show-cased (1), (2) was only shown for very similar tasks and it was unclear to what extent (2) is possible in a more realistic setting consisting of vastly different environments. MetaGenRL presents a novel approach to meta-RL that for the first time showcases both of these aspects: it outperforms hand-design algorithms such as PPO, REINFORCE and sometimes even DDPG; and it is able to generalize to vastly different environments.
>
> Regarding (1), while it is important to compare to DDPG it is equally important to consider the other baselines. In particular, while the meta-learned objective functions support learning rules similar to policy gradient estimators, they do **not** support DDPG in their current form. Indeed, during meta-testing there is no component that resembles DDPG in any way (a value function is only used as a constant input to the objective function). Based on this, one can not expect MetaGenRL to outperform DDPG, while on the other hand it is reasonable to expect it to be competitive with REINFORCE and PPO. As we show in the paper, MetaGenRL in fact greatly outperforms these algorithms, while DDPG is still better overall and remains a good target for future work that considers more expressive meta-learned objective functions.
>
> Regarding (2), it is also important to compare MetaGenRL to prior meta-RL approaches such as RL2 and EPG that were unable to showcase (2) to this extent. We find that RL2 overfits (which is a non-trivial observation) and that EPG is extremely sample inefficient.
>
> All in all we argue that MetaGenRL is an important step in realizing the potential of meta-RL.
>
> > “This suggests that this added complexity is not aiding in capacity and or hurting training.”
>
> Note that the learned loss does not modify a DDPG algorithm and there is only added complexity for meta-learning, not at meta-test time.
>
> > “This makes me fear there is something indirect and not interesting occurring …”
>
> As mentioned, MetaGenRL can not implement/modify DDPG, therefore something interesting must occur.
>
> > “Given how weak EPG is (as you stated for number of frames) and how RL^2 will never generalize across these different tasks it's hard to get a sense of the numbers.”
>
> As mentioned, it is important, fair, and meaningful to compare to EPG, and RL2. We updated the table to also include the comparison to PPO, DDPG, and on/off-policy REINFORCE from the appendix.
>
> > “I don't understand why meta-training on lunar and transferring to hopper does better than meta-training on hopper (table 1, middle column).”
>
> The table shows that meta-training on Lunar & Cheetah performed better in general on multiple test-time environments compared to meta-training on Cheetah & Hopper. It is difficult to precisely pinpoint how different combinations of environments affect meta-training.
>
> > “... I would appreciate if it put the meta-test performance on the same graph as meta-train performance …”
>
> Meta-training performance and (final) meta-test performance are not directly comparable. A population of sub-optimal agents (performance-wise) may already yield an objective function that can train an optimal agent from scratch. Note that while there are some fluctuations in meta-test performance, we observed an increasing trend on average as seen in figure 3 (now figure 4).
>
> > “How do you select when to test these algorithms?”
>
> We have tested the neural objective functions after 1 million timesteps of training per agent and this was not tuned in any way.
>
> > “Key details such as meta-training are also not discussed in depth nor ablated”
>
> Could you please elaborate on what aspects you find missing? It is our understanding that all meta-training details are available. Regarding ablations it is computationally infeasible to consider all possible variations, and we used our available resources on ablating the neural objective function, which we believe to be most important.
>
> [1/2, Continued in next reply]

---

> > ### Author Response · Authors · 2019-11-09
> > **Response to reviewer #2 with comments on improvements [2/2]**
> >
> > > “Hyperparameters of your baseline do not appear to be tuned (taken from appendix) where as for your method has a number of choices. How are you tuning these choices?”
> >
> > The DDPG baseline was derived from https://spinningup.openai.com/ (a tuned version on mujoco environments) and shares the same parameters with MetaGenRL where possible.  REINFORCE and PPO also use tuned parameters from the same source. We have not done an extensive hyperparameter search for MetaGenRL. We have validated the RL^2 parameters to work on a bandit-experiment from the original paper and derived parameters for the mujoco environments from the tuned configurations of https://ray.readthedocs.io/en/latest/rllib.html. For EPG we have used official code already tuned for mujoco benchmarks.
> >
> > > “Please include what error bars are for all plots.”
> > We believe only figure 4 (now figure 5) was missing this and corrected it.

---

> > > ### Comment · AnonReviewer2 · 2019-11-13
> > > **Question**
> > >
> > > Thank you for your in depth response! This has satisfied me and I plan on bumping my score up to a 6 -- weak accept.
> > >
> > > Judging from your comments I think I am misunderstanding your meta-test time procedure. I originally thought something like equation 6 was used for both meta-training AND meta-testing. I still have questions however. In particular, how is your algorithm not able to implement something like ddpg?
> > >
> > > You state -- " During evaluation (meta-test time), the meta-learned objective function can then be used to train a randomly initialized RL agent in a new environment."
> > >
> > > The learned objective function, L, is a function of trajectory, the current policy, and some value estimate.
> > >
> > > I believe your value function which is implemented by a Q function. I assume this Q function is learned from scratch when meta-testing as well though I couldn't find a reference to this. If L implements something akin to an identity function (and ignores all temporal aspects) I believe the resulting algorithm will look very similar to DDPG no?
> > >
> > > Would it be possible to include a meta-test time algorithm box as well?
> > >
> > > Putting this aside, why did you decide to have a different meta-training and meta-testing procedure? Both versions require collecting data, a replay buffer, training a Q function, and so on so it seems like they could do the same thing (equation 6 as opposed to directly minimizing L). Its not obvious to me that minimizing L will even do something reasonable as during meta-training L is only used to update Q (I believe?)
> > >
> > > =====
> > > Completely unrelated nit about your comment -- in particular (2). While I agree that the work showing transfer of RL algorithms across environments is interesting and under explored, there has been some instances in past work. This is also quite subjective -- what is a "different environment" anyway? First, https://arxiv.org/pdf/1812.01054.pdf shows transfer across atari games. https://s3-us-west-2.amazonaws.com/openai-assets/research-covers/retro-contest/gotta_learn_fast_report.pdf shows transfer across different levels using RL2 type approach. https://arxiv.org/pdf/1606.04671.pdf shows some transfer across atari games too. EPG (https://papers.nips.cc/paper/7785-evolved-policy-gradients.pdf) also tests very different tasks. I do believe your claim is technically true but its ignoring the fact that moving from (1) - (2) is continuous.

---

> > > > ### Author Response · Authors · 2019-11-14
> > > > **Second response to reviewer #2 with comments on improvements**
> > > >
> > > > Thank you for taking the time to discuss this further.
> > > >
> > > > During meta-test time we reinitialize the critic and the policy with random weights and only keep the weights of the objective function. Then we train in parallel: (1) A critic using the TD-error to estimate V. (2) The policy by following \nabla_\phi L_\alpha(\cdot).
> > > > The objective function is kept fixed during meta-test time.
> > > >
> > > > > “how is your algorithm not able to implement something like ddpg?”
> > > >
> > > > Similar to policy gradient methods \nabla_\phi V = 0, i.e. V is a constant w.r.t. to \phi. We do this by stopping the gradient. DDPG requires to backpropagate through the value function, thus L representing the identity function would not suffice. We will update the paper to clarify this distinction.
> > > >
> > > > > “during meta-training L is only used to update Q”
> > > >
> > > > This is a misunderstanding. L_\alpha is only used to update the policy \pi_\phi (‘learning’). Q_\theta is only updated by the TD-Error (equation 5). We then make use of Q_\theta to update L_\alpha (equation 6, which requires differentiating twice) **only during meta-train time** (‘meta-learning’). We have added an overview figure 1 to the updated submission that visualizes these different interactions.
> > > >
> > > > > “why did you decide to have a different meta-training and meta-testing procedure?”
> > > >
> > > > The meta-test procedure is designed to evaluate whether the objective function is able to train a randomly initialized agent from scratch. Hence, we only consider ‘learning’ by using the objective function, and prevent potential confounders that may arise when also simultaneously meta-learning.
> > > >
> > > > In contrast, during meta-training we also have ‘meta-learning’ due to updating the objective function. The analog to this is a researcher designing a new objective function (‘meta-learning’) and then using it to train RL agents (‘learning’).
> > > >
> > > > > “Related work”
> > > >
> > > > Thank you for pointing out this related work, we were aware of some of these already, and our goal is to incorporate all of these as part of a broader discussion (in the related work section) on generalization and transfer to other environments.
> > > >
> > > > > Paper update
> > > >
> > > > On Friday we will upload another updated version of our paper to incorporate your additional suggestions (such as the separate meta test-time box), while also incorporating the changes suggested by the other reviewers.

---

### Official Review · AnonReviewer3 · 2019-10-23
**Official Blind Review #3**

**Rating:** 6

**Review:**

The paper proposes to meta learn the objective function of a policy gradient algorithm using second order gradients of the objective function w.r.t the state-action value Q.

This is an interesting approach, however, I think the experimental evidence is not sufficiently convincing.

- In particular, I think the most important baseline that is compared against is not RL2, but DDPG: RL2 is not designed to generalize but to learn quickly on new tasks from the training-task distribution. Because the proposed algorithm does not depend on the observed states, it generalizes much better, but is also much slower than RL2. On the other hand, it shares a lot of design choices with DDPG: Using TD3 and Double Q-learning, as well as using as objective function the Q-values.
Looking at Figures 2, it is not clear that the proposed algorithm is substantially better than DDPG.
- Cheetah, Hopper and Lunar Lander are very simple environments. Evaluation on (slightly) larger scale environments would show that the algorithm can scale.
- The authors claim that the algorithm allows sharing of exploration strategies, which I don't believe can be the case based on it's current design.
- Lastly, I have a question about Fig 5a vs. Fig 2b: Shouldn't the performance of MetaGenRL be the same in both? It appears to perfom much better in Figure 2b.

Minor remark/question (didn't influence score):
The authors claim in the very first paragraph (and in the 4th paragraph) that inductive biases in humans are learned by natural evoluation through "distilling the collective learning experiences of many learners" by "learning from learning experiences". I'm not familiar with the relevant literature, but this seems like a strong statement which I believe should be supported by a citation.

Edit because I can't make my response visible to authors anymore:
Thank you for your response to my review and apologies for my delayed answer.

After reading your responses I agree that PPO is a fairer comparison than DDPG and that you are outperforming PPO is promising.
I further agree that it is relevant to show that RL2 overfits (although I personally don't find that very surprising - see below).

However, I still don't think that RL2 is a relevant baseline for this approach. There's a fundamental trade-off between the speed of adaptation and the amount of overfitting.
If I want to adapt very quickly (like RL2 does), I need to leverage as much task-information as possible, thereby overfitting to the task-distribution.
On the other hand, MetaGenRL is slow, it has training speed comparable with gradient-based approaches (by generalizes better by construction because it e.g. doesn't receive states as inputs).
Consequently, because MetaGenRL doesn't offer any speed improvements over gradient-based approaches, it should be compared to them, and not RL2.

Taken both the positive results vs. PPO and the negative results vs. DDPG, together with the fact that there's no learning speed advantage for MetaGenRL, I would see it as an interesting, and promising, research direction, but so far without proof that it can advance state of the art, as it looses to RL2 in terms of speed and DDPG in terms of final performance.

**Experience Assessment:**

I have read many papers in this area.

**Review Assessment: Checking Correctness Of Derivations And Theory:**

I assessed the sensibility of the derivations and theory.

**Review Assessment: Checking Correctness Of Experiments:**

I assessed the sensibility of the experiments.

**Review Assessment: Thoroughness In Paper Reading:**

N/A

---

> ### Author Response · Authors · 2019-11-09
> **Response to reviewer #3 with comments on improvements**
>
> Thank you for your review and valuable feedback!
>
> Before we proceed into detail we would first like to clarify some aspects that motivated this work, including our choice of experiments, and baselines.
>
> The premise of meta-RL is (1) that meta-learning learning rules for RL allows us to outperform existing human-engineered approaches on single environments, and (2) that learned learning rules are able to incorporate knowledge about learning in one task (or environment), to improve learning others. While prior work in meta-RL has show-cased (1), (2) was only shown for very similar tasks and it was unclear to what extent (2) is possible in a more realistic setting consisting of vastly different environments. MetaGenRL presents a novel approach to meta-RL that for the first time showcases both of these aspects: it outperforms hand-design algorithms such as PPO, REINFORCE and sometimes even DDPG; and it is able to generalize to vastly different environments.
>
> Regarding (1), while it is important to compare to DDPG it is equally important to consider the other baselines. In particular, while the meta-learned objective functions support learning rules similar to policy gradient estimators, they do **not** support DDPG in their current form. Indeed, during meta-testing there is no component that resembles DDPG in any way (a value function is only used as a constant input to the objective function). Based on this, one can not expect MetaGenRL to outperform DDPG, while on the other hand it is reasonable to expect it to be competitive with REINFORCE and PPO. As we show in the paper, MetaGenRL in fact greatly outperforms these algorithms, while DDPG is still better overall and remains a good target for future work that considers more expressive meta-learned objective functions.
>
> Regarding (2), it is also important to compare MetaGenRL to prior meta-RL approaches such as RL2 and EPG that were unable to showcase (2) to this extent. We find that RL2 overfits (which is a non-trivial observation) and that EPG is extremely sample inefficient.
>
> All in all we argue that MetaGenRL is an important step in realizing the potential of meta-RL.
>
> > “I think the most important baseline that is compared against is not RL2, but DDPG”
>
> As we discussed, it is very important to also compare to RL2 in the context of (2), and to consider the performance of MetaGenRL in relation to other baselines in the context of (1).
>
> > “Because the proposed algorithm does not depend on the observed states, it generalizes much better, but is also much slower than RL2”
>
> It is not clear that the poor generalization performance of RL2 is only due to conditioning on the state. In our experiments we observed that RL2 does not perform any meaningful learning during meta-testing (it has simply overfitted to the environment). Hence, while it is indeed “faster” in this regard this is mostly due to a limitation on RL2’s part. Nonetheless, MetaGenRL is able to outperform most of the time, even in those cases.
>
> > “Cheetah, Hopper and Lunar Lander are very simple environments. Evaluation on (slightly) larger scale environments would show that the algorithm can scale.”
>
> We agree that it would be interesting to consider more complex environments to test the limits of MetaGenRL. Nonetheless, the Mujoco simulator based environments are well established benchmarks in the context of (continuous control) (meta-)RL. It allows us to reuse hyperparameters, and compare more easily against other algorithms.
>
> > “The authors claim that the algorithm allows sharing of exploration strategies...”
>
> We agree that it is unlikely that the current architecture with the described inputs is able to learn an exploration scheme and we have not evaluated this empirically. The example in Section 3.3 was only meant for illustrative purposes, and we have removed this.
>
> > “Fig 5a vs. Fig 2b: Shouldn't the performance of MetaGenRL be the same in both?”
>
> Those results were obtained using a different seed for meta-training. In preliminary experiments we found that our results were consistent across different meta-training seeds (we already average over many agents), and so we have focused on different meta-testing seeds for computational reasons. That said, it would be better to also average over meta-training runs (yielding 6 * 6 = 36 configs), which we will soon add to the paper for Cheetah, Lunar -> Hopper to provide an indication of the variance also due to meta-training.

---

### Author Response · Authors · 2019-11-15
**Summary of changes after rebuttal**

We would like to thank the reviewers for their thoughtful reviews and useful feedback.

In the following we summarize the modifications that we have made to the paper since the start of the rebuttal.

* Fixed minor issues regarding the notation (R1)
* Improved the discussion of the LSTM as a general function approximator (R1)
* Updated Table 1 to include a comparison to the non-meta baseline results that was previously in the appendix (R2)
* Clarified why MetaGenRL can not implement DDPG, which motivates our comparison to PPO and the policy gradient baselines (R2)
* Clarified how we perform model selection (R2)
* Clarified how the baselines are tuned and the hyer-parameters are obtained (R2)
* Ensured that the captions describe what the error bars are for all plots (R2)
* Incorporated related work on transfer / generalization in RL (R2)
* Moved the algorithm box for MetaGenRL to the main text, and added a separate box for meta test time, to clarify the differences to meta training (R2)
* Added an overview figure to better explain the different interactions in MetaGenRL (R2)
* Removed the part on MetaGenRL meta-learning exploration strategies, which was only meant for illustration (R3)
* Investigated randomness during meta-training on the Cheetah and Lunar environments. While results from additional meta-training seeds did not alter our conclusions in any way, we do believe that it is valuable to also include additional meta-training seeds for all experiments in the future (see below) (R3)

In the future we will also

* Include an experiment, where MetaGenRL has been meta-trained on **many** different environments, to assess its capabilities (suggested by R1 & R3)
* Include additional seeds for meta-training for **all** experiments, to provide an indication of variance during meta-training (and meta-testing) in all settings.

Some of the reviewers' suggestions involved moving content from the Appendix to the main text, and we also added an additional figure. While this has slightly increased the length of the main content, we believe that the improved exposition makes this worthwhile.

---

### Author Response · Authors · 2020-02-14
**Summary of changes for the camera ready version**

Based on the area chair’s and reviewers’ feedback we have made the following additional changes to the camera ready version:

* We have re-ran all experiments to include 6 meta-train x 2 meta-test seeds (totaling 12 meta-test seeds) for MetaGenRL. In order to make this computationally feasible (as meta-training involves a population of agents) we have reduced the number of meta-training environment iterations from 1M to 600K, which gave similar results.
* For RL^2 these experiments now include additional environment interactions (from 50M to 100M) and also 6 meta-train x 2 meta-test seeds.
* For EPG these experiments now include up to 1 billion environment interactions per run (as opposed to 400M before), which forced us to consider 3 meta-train x 2 meta-test seeds (i.e. a total of 6 during meta-testing).
* We have added a new experiment in which we ablate the number of agents used in the population during meta-training MetaGenRL and confirm that a larger population is indeed beneficial.
* We have also added an experiment that investigates the benefit of meta-training on additional environments (with up to 40 agents), but were unable to report major improvements, possibly related to the current form of the objective function.
* While we previously reported that increasing the number of inner gradient updates from 1 to 3 was beneficial, we now find (after using additional seeds) that its performance is similar (with slightly larger variance) to using only a single gradient step. As before, we find that using 5 steps limits performance due to a large variance.
* We have incorporated several additional references to related work.

Finally, we emphasize that at this point all code to reproduce our results is available online.

---

### Decision · Program_Chairs · 2019-12-19

**Decision:**

Accept (Spotlight)

**Comment:**

This paper proposes a meta-RL algorithm that learns an objective function whose gradients can be used to efficiently train a learner on entirely new tasks from those seen during meta-training. Building off-policy gradient-based meta-RL methods is challenging, and had not been previously demonstrated. Further, the demonstrated generalization capabilities are a substantial improvement in capabilities over prior meta-learning methods. There are a couple related works that are quite relevant (and somewhat similar in methodology) and overlooked -- see [1,2]. Further, we strongly encourage the authors to run the method on multiple meta-training environments and to report results with more seeds, as promised. The contributions are significant and should be seen by the ICLR community. Hence, I recommend an oral presentation.

[1] Yu et al. One-Shot Imitation from Observing Humans via Domain-Adaptive Meta-Learning
[2] Sung et al. Meta-critic networks